# Heat Stress Reduces the Susceptibility of *Caenorhabditis elegans* to Orsay Virus Infection

**DOI:** 10.3390/genes12081161

**Published:** 2021-07-28

**Authors:** Yuqing Huang, Mark G. Sterken, Koen van Zwet, Lisa van Sluijs, Gorben P. Pijlman, Jan E. Kammenga

**Affiliations:** 1Laboratory of Nematology, Wageningen University & Research, Droevendaalsesteeg 1, 6708 PB Wageningen, The Netherlands; yuqing.huang@wur.nl (Y.H.); mark.sterken@wur.nl (M.G.S.); koen.vanzwet@wur.nl (K.v.Z.); lisa.vansluijs@wur.nl (L.v.S.); 2Laboratory of Virology, Wageningen University & Research, Droevendaalsesteeg 1, 6708 PB Wageningen, The Netherlands; gorben.pijlman@wur.nl

**Keywords:** *C. elegans*, Orsay virus, heat stress

## Abstract

The nematode *Caenorhabditis elegans* has been a versatile model for understanding the molecular responses to abiotic stress and pathogens. In particular, the response to heat stress and virus infection has been studied in detail. The Orsay virus (OrV) is a natural virus of *C. elegans* and infection leads to intracellular infection and proteostatic stress, which activates the intracellular pathogen response (IPR). IPR related gene expression is regulated by the genes *pals-22* and *pals-25,* which also control thermotolerance and immunity against other natural pathogens. So far, we have a limited understanding of the molecular responses upon the combined exposure to heat stress and virus infection. We test the hypothesis that the response of *C. elegans* to OrV infection and heat stress are co-regulated and may affect each other. We conducted a combined heat-stress-virus infection assay and found that after applying heat stress, the susceptibility of *C. elegans* to OrV was decreased. This difference was found across different wild types of *C. elegans*. Transcriptome analysis revealed a list of potential candidate genes associated with heat stress and OrV infection. Subsequent mutant screens suggest that *pals-22* provides a link between viral response and heat stress, leading to enhanced OrV tolerance of *C. elegans* after heat stress.

## 1. Introduction

*Caenorhabditis elegans* is a free-living bacterivorous nematode, and natural populations are closely associated with decaying organic matter. This leads to a continuous exposure to many different (a)biotic factors, including pathogens [1]. Abiotic factors include ambient temperature, moisture conditions and osmotic changes among many others. Pathogens include a range of bacteria, fungi, microsporidia, and viruses [2]. Many of these abiotic and biotic challenges disturb proteostasis and trigger intrinsic stress responses. Maintaining proteostasis is essential for survival, and multiple stress response pathways are involved in protecting *C. elegans* from these negative stress effects. In the laboratory, *C. elegans* has served as a model species to understand the molecular responses of abiotic stress and pathogens, which is facilitated by its completely sequenced and annotated genome, genetic tractability, transparent body, ease of cultivation in the lab, and its relatively short developmental period [3]. Different pathogens affecting *C. elegans* have been studied in the laboratory, including microsporidia, bacteria, and a virus [1]. Among these pathogens, the Orsay virus (OrV) is currently the only virus known that naturally infects *C. elegans*. OrV infection triggers three antiviral defense mechanisms: the RNA interference (RNAi) response, the uridylation responses, and the Integrated Pathogen Response (IPR), an innate transcriptional response [4]. The IPR pathway is regulated by the antagonistic *pals-22* and *pals-25* genes, members of a family of divergent genes defined by the presence of an ALS2CR12 domain. *pals-22* is a repressor of the IPR pathway, whereas *pals-25* is a positive regulator [5,6].

In addition to viral infection as a well-studied biotic factor, the response to abiotic factor heat stress (HS) has been thoroughly investigated in *C. elegans*. HS refers to the temperature conditions of the ambient environment exceeding the optimal range of an organism, which might lead to the perturbation of cellular function thus causing protein damage and aggregation and the formation of toxic protein oligomers [7,8]. To combat the effect of HS, processes like autophagy and the heat-stress response (HSR) maintain proteostasis [5,9]. HS induces the HSR, including the activation of the heat shock factor HSF-1 to form oligomers. Once HSF-1 is oligomerized, it translocates to the nucleus and activates the HSR, after which these HSR proteins prevent the formation of misfolded protein oligomers and help to refold misfolded proteins [10,11].

The IPR and HSR share particular protective functions, namely both processes protect *C. elegans* from proteostatic stress. This can be illustrated by the loss-of-function mutation of the IPR repressor *pals-22* that leads to enhanced thermotolerance and increased resistance against natural intracellular pathogens [5,6]. Given that the IPR and HSR have these gene functionalities in common, we hypothesize that a shared mechanism or link might be present. To gain more insight into this potential link, we combined OrV infection with HS exposure in *C. elegans*. The goal was to investigate whether HS influences the OrV susceptibility of *C. elegans*. We studied the most widely used reference strain, wild-type Bristol N2 (N2), the natural OrV sensitive strain, wild-type JU1580, in which OrV was originally found [12] and wild-types JU1511 and CB4856. Our results show that HS reduces the susceptibility of *C. elegans* to OrV infection and suggest that *pals-22* plays an important role in this process. The outcomes suggest that the effect of HS on viral sensitivity may differ across different wild type genetic backgrounds.

## 2. Materials and Methods

### 2.1. Nematode Culturing and Strains

Hermaphrodites of *C. elegans* strains N2, JU1580, CB4856, and JU1511 were kept under standard conditions at 20 °C on Nematode Growth Medium (NGM) seeded with *Escherichia coli* OP50 as a food resource. For stage synchronizing, starved worm populations were bleached with a mixture of NaOH, Milli Q, and bleach, and the eggs were then transferred to fresh 9 cm 2x NGM (double density of agar) plates. Worm eggs were incubated for 20 h at 20 °C until the L1 stage [13].

Mutant strains RB1330 *npr-1* (*ok1447*), VC3467 *hsp-1* (*ok1371*), RB1099 *hsp12.6* (*ok1077*), and RB791 *hsp-16.48* (*ok577*) were ordered from the Caenorhabditis Genetics Center (CGC), and RB791 and RB1330 were then backcrossed with our laboratory strain N2 for six generations to replace the genome background with our N2 by more than 99%. Genotypes were checked with PCR (Appendix A) using primers suggested by the CGC. ERT356 *pals-22* (*jy1*) and ERT463 *pals-22 pals-25* (*jy1jy11*) mutants were previously constructed using an EMS screen [6].

### 2.2. OrV Infection and HS Experiment

The experimental setup is shown in Figure 1a. OrV stock was obtained by incubating previously infected JU1580 populations [12]. By means of a viral dose–response curve, we selected the efficient volume 50 µL virus stock in a per 500 µL infection mixture (the lowest volume needed to reach the maximum viral replication). During all OrV infection experiments, 9 cm 2× NGM plates were used to incubate the worms. For collection, there were about 400 L1 stage worms on each plate.

For infecting *C. elegans*, a liquid infection assay was applied using L1 stage worms [13]. Briefly, plates were rinsed from the plates with M9 buffer, collected in Eppendorf tubes, and centrifuged for a short time to pellet the worms. Thereafter the buffer was removed, and 500 µL of infection solution (350 µL of M9, 50 µL of virus stock, and 100 µL of OP50 in luria broth (LB)) or mock solution (400 µL of M9 and 100 µL of OP50 in LB) was added. The worms were incubated in infection solution for 1 h in Eppendorf tubes at room temperature and were regularly mixed to infect them with OrV. Next, the worms were pelleted through centrifugation, and after washing thrice with M9, the supernatant was removed ([14], p. 58) Worms were spread to fresh plates for further incubation.

To address the effect of HS on OrV infection, we conducted an experiment exposing *C. elegans* to both stressors. Because the order of the HS applications might influence the viral load, HS was applied 3 h before OrV infection. In the second treatment, HS was applied at 3 h post OrV infection (Figure 1a). OrV infections without HS treatment were included as controls.

For the HS experiment, the temperature was set to 35 °C for 2 h in a climate cabinet (Elbanton) because it would trigger the HSR without killing the worm [15], which is necessary for later OrV infection. First, bleached eggs were hatched at 20 °C and grown for 20 h. Subsequently, for the control treatment, the OrV infection started after 23 h. For the HS–OrV treatment, from 20 h, the worms were grown at HS (35 °C for 2 h), after which they were allowed to recover (20 °C for 1 h). After that, the OrV started at the same time as the control at 23 h. For the OrV–HS treatment, the OrV started at 23 h, and the HS recovery phase was then applied. Subsequently, 30 h after OrV infection (L4 stage to early adult), the samples were collected by flash freezing the worms washed with M9 buffer on NGM plates using liquid nitrogen. Samples were stored at −80 °C until RNA isolation, cDNA synthesis, and RT-qPCR analysis.

For the N2 and JU1580 experiments, nine biological replicates were applied. For the N2 and JU1511 experiments, seven biological replicates were applied. For the mutant experiments and gene expression experiments, seven biological replicates were applied. During the experimental process, two technical replicates per strain were applied, and before freezing, the two replicates were combined. For all viral experiments, only infected replicates were displayed in the results because we are not able to rule out the possibility that a lack of infection may stem from technical errors.

### 2.3. Microarray Experiment

To explore the potential genes associated with lower OrV susceptibility after HS, the global gene expression profile was measured using microarray analysis. Gene expression was measured in the strains N2 and CB4856, as these genotypes have been well-studied in regard to their gene-expression responses to temperature changes and HS [15,16,17]. Although the microarrays were originally designed for the N2 strain, they can be used for the CB4856 strain [18,19]. Additionally, we chose CB4856 and not JU1580 because the latter genotype is not as well-characterized as CB4856, and therefore, we have less insight into differential hybridization effects based on genotype [18]. For the microarray experiment, the worms were treated as shown in Appendix A, with HS at 46 h (if HS is involved in the treatment), OrV infection at 50 h, and sampling at 80 h. For the mock treatment, a mock-infection solution was used instead of the virus stock. This mock-infection solution was prepared in the same manner as the virus stock, with the difference being that the lysed nematodes (strain JU1580) were healthy instead of infected.

N2 and CB4856 were each fitted to four microarrays, with four treatments for each strain. Each strain had a sample that was treated with both HS and OrV infection, one with HS and mock-infected, one that was OrV infected only, and one that only underwent a mock infection.

### 2.4. Microarray Labelling and Scanning

The microarray experiment was conducted as described previously [20]. In short, the ‘Two-Color Microarray-Based Gene Expression Analysis; Low Input Quick Amp Labeling’ protocol, version 6.0 from Agilent (Agilent Technologies, Santa Clara, CA, USA) was used. For measuring expression, the *C. elegans* (V2) Gene Expression Microarray 4 × 44k chips were used (Agilent). Scanning was done with an Agilent High Resolution C Scanner. For extraction, we used Feature Extract (v. 10.7.1.1).

### 2.5. Microarray Data Processing

The extracted intensities were processed as recommended and using the Loess method for within-array normalization and the Quantile method for between-array normalization [21,22]. The values were log_2_-transformed, and a ratio with the mean was also calculated based on formula:*R*_[*i*,*j*]_*=* [log]_2_ (*y_i_*_,*j*_/(*mean y_i_*))
where *R* is the log_2_ relative expression of spot *i* (*i* = 1, 2, …, 45,220) for sample *j*, and *y* is the intensity (not the log_2_-transformed intensity) of spot *i* in sample *j*.

### 2.6. Microarray Data Analysis

Only a single sample was available for each treatment because the microarray data were used for exploratory purposes. Therefore, a combination of correlation analysis and principal component analysis was used to understand the consistency of the data. Correlation analysis (Pearson) and principal component analysis were conducted on the normalized log_2_ (of the raw intensity data) relative expression using *cor* and *prcomp* (with scale. = TRUE) in “R”.

To identify affected genes, gene expression fold-changes were calculated for the comparisons of N2–control–mock versus N2–control–infected, N2–heat stress–mock versus N2–heat stress–infected, CB4856–control–mock versus CB4856–control–infected, and CB4856–heat stress–mock versus CB4856–heat stress–infected. Thereafter, we took the median FC over all of the spots detecting the same gene. These were compared at the cut-off of |FC| > 1 and specifically for genes known to be involved in (a)biotic stress, including *hsp*- and *pals*- genes. The OrV-affected genes were selected from [23], and the heat-stress involved genes were selected from [20].

### 2.7. Food Intake Assay

To investigate whether food intake during the assay influences the viral load, we conducted two experiments with *C. elegans* N2. In Experiment 1, we measured food intake during HS, and in Experiment 2, we measured food intake after HS. This experiment used 6 cm 1x NGM plates. The experimental setup is shown in Figure 2a.

Experiment 1: Worms were synchronized by means of bleaching and were allowed to grow for 23 h (L1 stage), after which they were transferred to plates containing red fluorescent beads (Sigma L3280, red fluorescence) [24,25], to which they were exposed at 35 °C for 2 h. Control worms were exposed at 20° C for 2 h. Directly after that, the fluorescent signal was visualized and calculated using a microscope (Zeiss Axio Observer Z1 inverted microscope, magnification 40×) and the software ImageJ [26,27]. The food index was calculated as the number of pixels showing a fluorescent signal, which was normalized to the size of the whole worm body.

Experiment 2: Worms were synchronized by means of bleaching and were allowed to grow for 23 h, (L1 stage) after which they were transferred to plates containing fluorescent beads [24] to which they were exposed at 35 °C for 2 h. Control worms were exposed at 20 °C for 2 h. After that, the worms were transferred to plates containing fluorescent beads and were incubated at 20 °C for 1 h, which is the same temperature and incubation time required for OrV infection. After that, the fluorescent signal was visualized and was calculated using a microscope (Zeiss Axio Observer Z1 inverted microscope, magnification 40×) and the software ImageJ [26,27]. The food index calculation is the same as Experiment 1.

For both experiments, there was a negative control (same treatment but plates without fluorescent beads), and eight replicates each representing a single worm were applied, using four worms from two plates under the same treatment (plates containing fluorescent beads).

### 2.8. RNA Isolation

The RNA isolation was performed with a Maxwell^®^ AS2000 (Promega, Madison, WI, USA) using the Maxwell^®^ 16 LEV plant RNA kit (Promega, Madison, WI, USA) on the previously frozen samples. A small modification was made to the lysis step, adding a proteinase K digestion [20]. After isolation, the concentration was measured using the Nanodrop (Thermofisher, Waltham, MA USA). RNA samples were stored at −80 °C.

### 2.9. RT-qPCR and Data Analysis

Bio-Rad IQ5 was applied for both the *pals* gene expression test and the viral test. Genes Y37E3.8 and *rpl-6* were used as housekeeping reference genes. RNA1 and RNA2 were amplified for the detection of viral RNA [13] (primers see Appendix A).

After the Ct value was measured, the relative expression was calculated according to [13]:$$E=\frac{Q_{v}}{0.5*\left(\left(Q_{rpl-6}/{\bar{Q}}_{rpl-6}\right)+\left(Q_{Y37E3.8}/{\bar{Q}}_{Y37.3.8}\right)\right)}$$
in this formula, *E* represents relative expression, *Q* is the transformed expression, and *v* indicates one of the target genes (viral genes/*pals* genes). The expression of the target genes was normalized to the household genes *rpl-6* and Y37E3.8 for further comparison. Comparisons were tested by means of the Tukey multiple comparisons test (confidence level = 0.95).

## 3. Results

### 3.1. Susceptibility of C. elegans to OrV before and after HS

The HS and OrV treatments were applied to both N2 and the OrV sensitive strain JU1580 (Figure 1b). Under control conditions (only OrV infection), JU1580 was indeed more susceptible than N2 (Tukey HSD test, *p* < 0.01) [13]. For JU1580 as well as N2, it was observed that for both HS before OrV infection and for HS after OrV infection, the viral load was decreased compared to the control (Tukey HSD test, *p* < 0.01) (Figure 1b). For all treatments, JU1580 was more susceptible than N2. In sum, the OrV infection levels were lower for both N2 and JU1580 when HS was applied. These results illustrate that both N2 and JU1580 became less susceptible to OrV when HS was applied right before or after infection.

To further investigate the effects of a virus and HS in *C. elegans*, we extended our study to another wild type, JU1511. We exposed N2 and JU1511 to OrV at two different ages, 23 h and 50 h, as well as to HS before 50 h infection and after 23 h infection (Appendix Aa). We found that viral load in N2 at 50 h and 23 h (Con-OrV1 and Con-OrV2) (Appendix A) were not significantly different and replicable (i.e., the same as in Figure 1). However, HS before and after infection did not result in a lower viral load. The viral load in JU1511 under HS before OrV significantly increased compared to the control (Con-OrV1) (Tukey HSD test, *p* < 0.05), while no difference was found for N2 under the same treatment (Tukey HSD test, *p* > 0.05). In sum, the OrV sensitivity of *C. elegans* can be affected by HS, and the effect of HS on viral sensitivity may differ across different wild type genetic backgrounds and the age of the worm when exposed to HS.

Since the OrV load may potentially be influenced by differences in food intake during and/or after HS or due to the molecular stress-response in general, we next conducted a food intake experiment.

### 3.2. HS Does Not Influence the Food Intake of C. elegans

*C. elegans* is infected by OrV via ingestion through the digestive gut system [28]. Hence, it could be that the ingestion of food was affected by the HS, influencing the viral exposure. We tested if HS influences the bacterial food intake during and after HS relative to a control of 20 °C in the N2 strain (Figure 2a).

It was found that after and during HS, the food intake by the worms as measured by the ingestion of fluorescent beads was not affected relative to the respective controls (Tukey HSD test, *p* > 0.05) (Figure 2b). From this experiment we concluded that the initial virus levels inside the worm were not affected by differential food intake, indicating that the change of viral load in Figure 1 was not caused by heat induced changes in bacterial food intake, but by mechanisms downstream of viral intake.

### 3.3. Candidate Genes Selected Underlying Combined Viral Sensitivity and HS

We exposed N2 and CB4856 to four treatments: (i) HS exposed and infected (HS OrV), (ii) mock-infected and HS exposed (mock HS), (iii) infected at 20 °C (‘control’, CT OrV), and (iv) mock-infected at 20 °C (mock CT). As it was an exploratory experiment with the goal of determining candidate genes, only one biological replicate was tested, but it should be noted here that four technical replicates were spotted for all of the transcripts. We first tested whether the data were structured as expected based on literature. Namely, the HS effect was expected to be larger than the strain effect, and the strain effect was expected to be larger than the effect of infection with Orsay virus [15,23,29]. Indeed, by using correlation analysis and principal component analysis, we saw that the data clustered as expected (Appendix Aa,b). This shows that the measurements of a single microarray experiment are robust.

We found clear expression differences between OrV-infected and mock-infected samples measured in N2 and CB4856 (Figure 3a). Subsequently, we calculated fold-changes in the gene expression of OrV-infected versus mock-infected samples (Appendix A) and focused our analysis on the genes known to be involved in the HSR and OrV infection, including the HSP and IPR genes (Appendix A) [20,23]. We found that for 59 genes, expression changes were seen when comparing differential expression after OrV exposure. In CB4856, a subset of genes was upregulated upon OrV exposure, but only when the worm was heat stressed, and a second subset of genes were upregulated upon OrV exposure, independent of heat stress, while in N2, there were no genes whose upregulation upon OrV infection only occurred when the worm was heat stressed (Figure 3b). Furthermore, we confirmed that our experiment detected the differential expression of many genes that had previously been associated with OrV infection in N2 [20,23].

We investigated the overlapping genes differentially expressed in both genotypes to increase the chance that the genes were not differentially expressed by chance. Notably, this group of 59 genes contained 13 *pals*-genes and 7 serpentine receptor genes (Appendix A). Although none of the *hsp*- genes were found in the overlapping group, we did notice that some expression differences could be seen when looking at the genotypes separately, but these were generally more prominent in CB4856 (Figure 3b).

### 3.4. HS Effect on Viral Sensitivity of the pals-22 and the pals-22 pals-25 Mutants

Although we did not find *pals-22* and *pals-25* in the overlapping genes differentially expressed in both N2 and CB4856, *pals-25* was significantly upregulated, and *pals-22* was significantly down regulated in N2 (Figure 3a). Therefore, we continued to investigate *pals-22* and *pals-25* in N2. Knockout mutants *in pals-22* displayed an increased thermotolerance [5]. Since *pals-25* acts antagonistically to *pals-22,* we reasoned that *pals-22* could be a candidate gene involved in HS as well as in viral responses and that the *pals-22* effect could be neutralized by *pals-25*. The same experimental treatments as those applied to the wild types (Figure 1) were applied as described above, this time using *pals-22* mutant and *pals-22 pals-25* loss-of-function double mutants (Figure 4).

It was confirmed that the *pals-22* single mutant was less susceptible than N2 under control conditions (Con-OrV) [6] (Tukey HSD test, *p* < 0.05). We found that the level of the viral load of the double mutant was not significantly different compared to N2 under any of the treatment conditions, the double mutant showed decreased infection when the strain was also treated with HS after OrV. HS does not cause a significant decrease in viral load for the *pals-22* mutant compared to N2. This suggests that the sensitivity of *pals-22* mutant to OrV infection is hardly affected by HS.

As we also wanted to explore more subtle transcriptional effects, we selected *hsp-12.6* and *hsp-1* mutants since these genes showed differential expression depending on HS application (Figure 3b). The *hsp-16.48* mutant was selected because *hsp-16.41* and *hsp-16.2* mutants were not available, and *hsp-16.48* one closest to the mutation that was available. As a control, we included *npr-1* because it was involved in both interactions with bacteria and oxidative stress (pathogen and abiotic factor) [30,31,32]. The mutants were exposed to the same experiment as described in Figure 1a. We included JU1580 as a reference as well.

The mutants were tested using the same conditions that resulted in significant results in wild type N2 (Figure 5). For all of the mutants, we found a similar pattern compared to N2, and they did not differ in viral load compared to N2 for all treatments (Tukey HSD test, *p* > 0.05). Again, JU1580 was found to be more susceptible to OrV compared to N2 under control conditions (Tukey HSD test, *p* < 0.05). We conclude from this experiment that *hsp-12.6*, *hsp-1, hsp-16.48*, and *npr-1* were not involved in either the HS response or the IPR pathway.

To assess the role of *pals-22, pals-25, pals-6* and *pals-14* in JU1580, we measured their expression in wild type JU1580 (Figure 6). Since the same level of reduction of viral load appeared under both two experimental treatments (HS–OrV, OrV–HS), one treatment, HS–OrV, was chosen from these two for gene expression detection. The expression of the *pals* genes was not significantly changed upon HS (Tukey HSD test, *p* > 0.05).

## 4. Discussion

### 4.1. HS Affects OrV Viral Load in C. elegans

The effect of HS and virus infection has extensively been investigated in *C. elegans*. However, so far, insight into how HS affects viral infection is very limited. It was found earlier that mutants in the IPR gene *pals-22* displayed thermo tolerance and increased viral susceptibility, but this study did not look into the interaction between the two types of biotic and abiotic stressors [6]. Since host–pathogen–environment interactions are essential in understanding disease emergence and spread [33], experiments that combine different stress factors will contribute to a more refined insight into the potential mechanisms of the interaction. However, combinatory stress experiments are complex, and many aspects come into play when it comes to designing such experiments. For instance, the timing, level, and order of stress make a big difference. Here, we chose to take a first step into the combinatory approach of HS and virus infection by applying HS before and after viral infection. For the wild types Bristol N2 and JU1580, the viral load significantly decreased under HS right before/after OrV infection. For N2, a HS treatment 27 h after exposure to OrV did not result in a significant change in the viral load. At 27 h after OrV infection, viral RNA replication may have already reached its maximum level ([14], p. 63), and therefore, the HS response would have no effect on the viral loads anymore. Overall, the results imply that HS before and directly after OrV infection increases the ability of *C. elegans* to combat the viral infection, leading to a reduced susceptibility. The exact mechanism is still unknown, but when HS is applied prior to the virus, HS may affect virus infection at the level of entry or viral RNA replication. It should be noted that during HS, we noticed that the worms grew a bit slower than the worms kept at 20 °C. In this case, we wondered whether developmental delay would influence the viral load. Previous work ([14], p. 63) reported that for N2, the larval stage does not influence the maximum viral load. In our experiment, the worms were infected at 23 h, while the developmental delay only occurred at the L1 to L2 stages [14]. Therefore, the viral load difference measured in our experiments was not likely to be caused by developmental difference.

### 4.2. HS Effects on OrV Viral Load May Depend on the Genetic Background

By comparing viral susceptibility in three different backgrounds, Bristol N2, JU1580, and JU1511, we found that the genetic background may influence the interaction effect of HS and viral infection. It is well known that a different genetic background can have strong differential effects on complex phenotypes [34,35] and that even the phenotypic effects of strong single gene mutations can be modulated. Our results suggest that the genetic background may also have genetic modifiers that can modulate the effect of viral infection in combination with HS. Analyzing the differential polymorphic regions related to IPR and HSR may be a step forward in identifying the potential background modifiers. We found that under HS conditions, JU1580 viral loads dropped to similar levels as N2, suggesting that a common allelic mechanism underlies this response for both strains in a similar way. As previous studies also revealed, the widely studied canonical strain Bristol N2 is more resistant against OrV infection compared to the sensitive strain JU1580. A lack of antiviral small RNA causes the difference between the two strains [18], yet the effect of temperature does not lead to a different viral load effect. In this case, we speculate that the HSR might interact with the viral response pathway. Genetic background effects were further substantiated by the microarray data. A subset of genes was upregulated in CB4856 after OrV exposure and HS and a second subset of genes were upregulated upon OrV exposure, independent of HS. This contrasts with N2, for which we did not detect genes whose upregulation upon OrV infection only occurred when the worm was heat stressed.

### 4.3. Col Genes Play a Role in HS and OrV Effects

The microarray results showed that genes from the collagen (*col*) family were strongly affected (Appendix A) upon the induction of HS. *Col* genes are involved in nematode cuticle structure, which is critical for protection and locomotion during and after HS. Jovic et al. (2017) conducted a high-resolution time series of increasing HS exposures and studied transcriptional patterns in *C. elegans*. *Col* genes were also substantially affected [20]. The expression of the genes belonging to the heat-shock protein family were only mildly affected by the HS in our experiment. Genes coding for heat-shock proteins have been shown to respond very quickly after a severe bout of heat. After heat stress, the induction quickly fades away, and this is also reflected in Appendix A. Regarding the microarray candidate gene detection and the *pals* gene expression experiment, it should be noted that the expression of HSR-related genes is highly dynamic [20]. Since we measured gene expression at one particular time point, it could well be that if gene expression had been conducted at a different age, we would have detected other genes that we could test for the mutant screens. For instance, HSR gene expression declines rapidly at a certain age, for example, at the early adulthood stage, stress responses will be repressed [8]. Thus, a more dynamic analysis could be an option to obtain a more precise and complete insight.

### 4.4. pals-22 May Play a Role in Combined HS and OrV Exposure

Under control conditions, i.e., only OrV infection, the *pals-22* mutant was less susceptible to OrV infection compared to the *pals-22/25* double mutant. This corresponds with the fact that both genes work antagonistically, yielding wildtype phenotypic stress responses in the double mutant [6]. Moreover, the viral sensitivity of the *pals-22* mutant was not influenced by HS, while the *pals-22/25* mutant behaves similarly to N2. We suppose that this is because the knockout of the *pals-22* gene removes its repression effect in the IPR response, which might lead to a state where the innate immune response might have reached its maximum capacity to improve viral resistance, and therefore, HSR no longer changes. In this case *pals-22* could be considered as a candidate gene that is involved in the IPR and HSR pathways. Alternatively, it could be noted that the *pals-22* mutant does not have the increased tolerance against OrV caused by HSR. From previous research, the *pals-22* mutant has increased thermotolerance on long-term HS [6].

To test if HS affected the IPR genes in JU1580, we also measured *pals* gene expression in the JU1580 strain after HS. In JU1580, the *drh-1* gene is not functional, and the IPR is not activated upon infection [36], but it could be that the IPR pathway in JU1580 becomes activated after HS, thus explaining the lower viral susceptibility (Figure 6). We found that none of the *pals* genes showed a significant change under the HS. This suggests that the IPR pathway is not activated by HS.

## 5. Conclusions

Overall, our results point at the fact that HS could enhance the tolerance of *C. elegans* against OrV infection, depending on the time after infection. The food intake assay showed that the difference was caused by host innate responses. However, the four candidate genes selected were confirmed not to be involved in the interaction between IPR and HSR. We advocate that a more dynamic analysis at different time points could provide more detailed insights. *pals-22* could be considered as a candidate gene involved in the two pathways. In natural populations, *C. elegans* is exposed to a multitude of biotic and abiotic factors, some of which trigger a specific innate response of *C. elegans*. We speculate that the underlying response pathways might interact or are intertwined with each other. As previously reported, increased temperature at an early stage induces lasting immunity to bacterial infection in *C. elegans* [37]. From our study, we found that temperature influences the result of virus infection. This may imply that temperature might also influence the responses to other pathogenic microbiota that trigger the same response pathway, such as microsporidia [4]. The pathogenic responses might be influenced by other abiotic factors, such as oxygen. Studying combined biotic and abiotic factors would provide more insight into the potential interaction of the response pathways, and the dynamic of organism evolution in more complicated environments.

## Figures and Tables

**Figure 1 genes-12-01161-f001:**
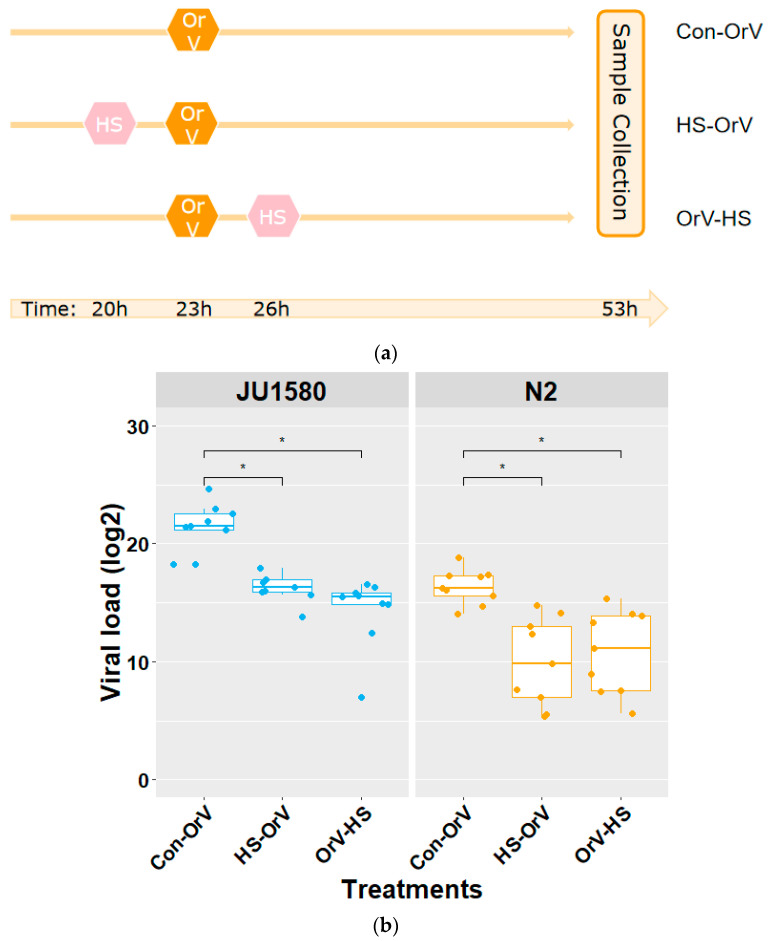
Viral load of JU1580 and N2 under combined HS and OrV infection (**a**) and timeline of the three experimental treatments. (**b**) The viral load of JU1580 and N2 under heat stress (HS). Con-OrV: Orsay virus infection; HS-OrV: HS before Orsay virus infection; OrV-HS: HS after Orsay virus infection. Dots: infected biological replicates. *: significantly different, Tukey HSD test, *p* < 0.05.

**Figure 2 genes-12-01161-f002:**
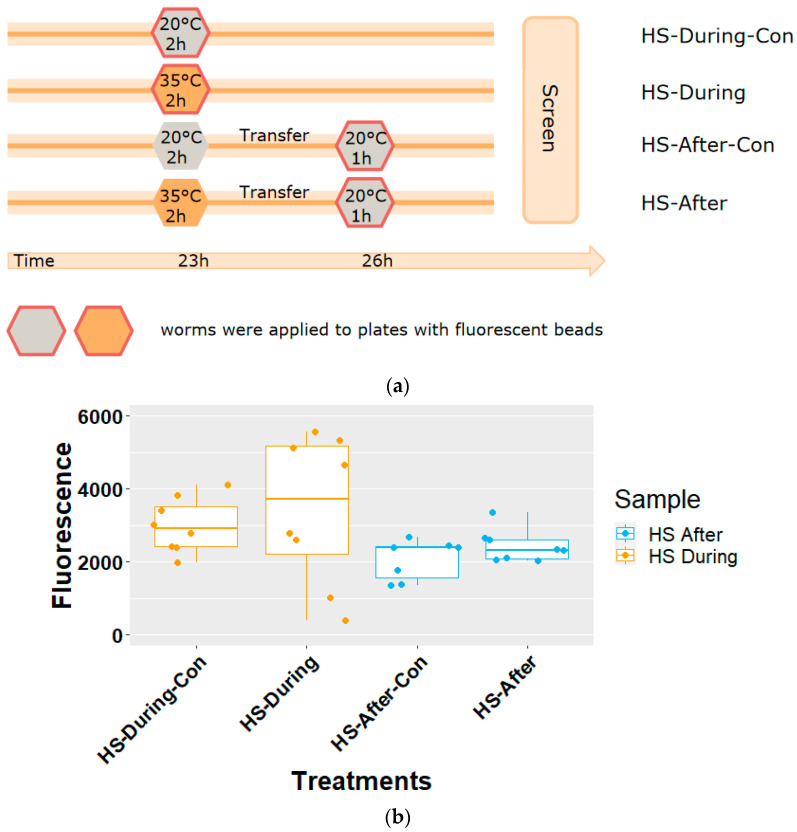
Food intake assay of *C. elegans* N2 during and after HS. (**a**) Timeline of food intake assay. (**b**) Assay result. HS-After: food intake after heat stress; HS-After-Con: control of HS-After; HS-During: food intake during heat stress; HS-During-Con: control of HS-During. Fluorescence: normalized fluorescence in pixel units. Dots: biological replicates.

**Figure 3 genes-12-01161-f003:**
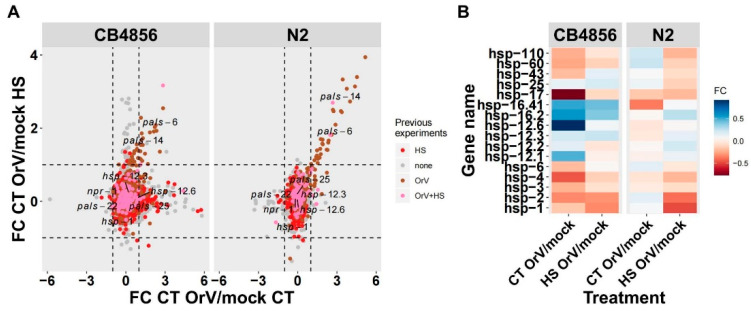
Impact of HS on OrV-induced gene-expression. (**A**) The log_2_ fold-change (FC) expression ratio between OrV-infected and mock-infected samples measured in CB4856 and N2. For each treatment-strain, combination *n* = 1 was used and measured by microarray. Each dot represents a gene and was given a color coding when detected in OrV literature (brown), HS experiments (red), both (pink), or none (grey). Labels of gene-names were added based on experiments presented in this paper. The dashed lines indicate a cut-off of |FC| > 1. (**B**) A heat-map of the FC expression differences of the *hsp*-genes. Blue indicates higher expression upon OrV-infection, and red indicates decreased expression.

**Figure 4 genes-12-01161-f004:**
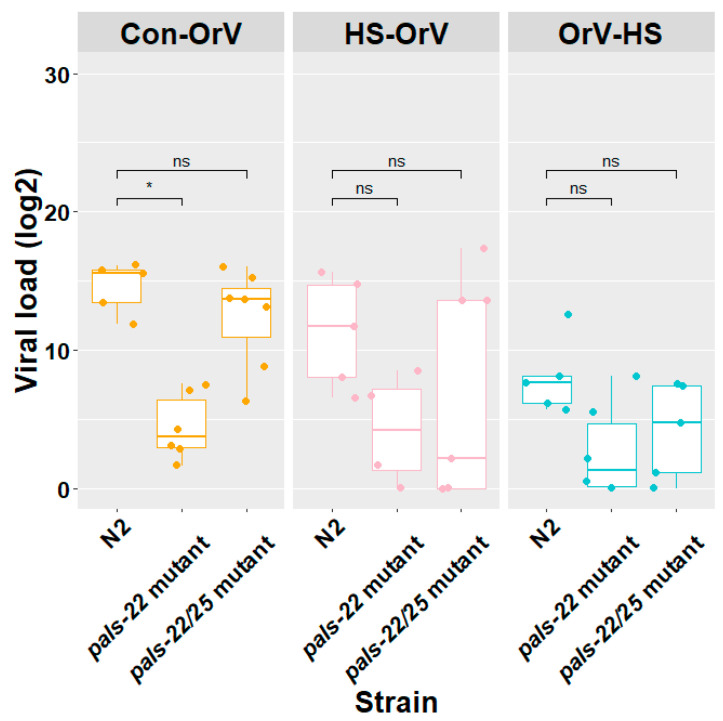
Viral load of N2, the *pals-22* and the *pals-22 pals-25* mutants under combined HS and viral stress. Con-OrV: control with only OrV infection; HS-OrV: HS before Orsay virus infection; OrV-HS: HS 2 h after Orsay virus infection. Dots: infected biological replicates (each dot is a biological replicate). Differences with N2 were tested, Tukey HSD test. *: significantly different, *p* < 0.05.

**Figure 5 genes-12-01161-f005:**
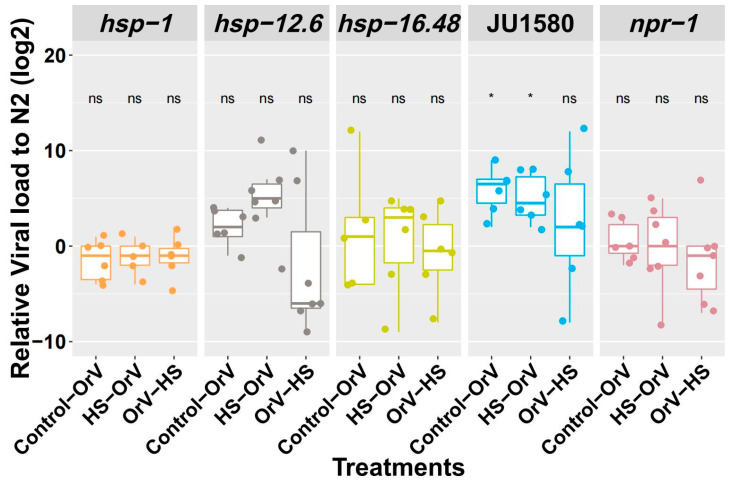
Viral load detection on JU1580 and mutants of *hsp-1, npr-1, hsp-12.6,* and *hsp-16.48*. The y axis represents the viral load difference from N2 within the same batch. Dots: infected biological replicates (each dot is a biological replicate). Differences with N2 were tested. Tukey HSD test, *: significantly different, *p* < 0.05.

**Figure 6 genes-12-01161-f006:**
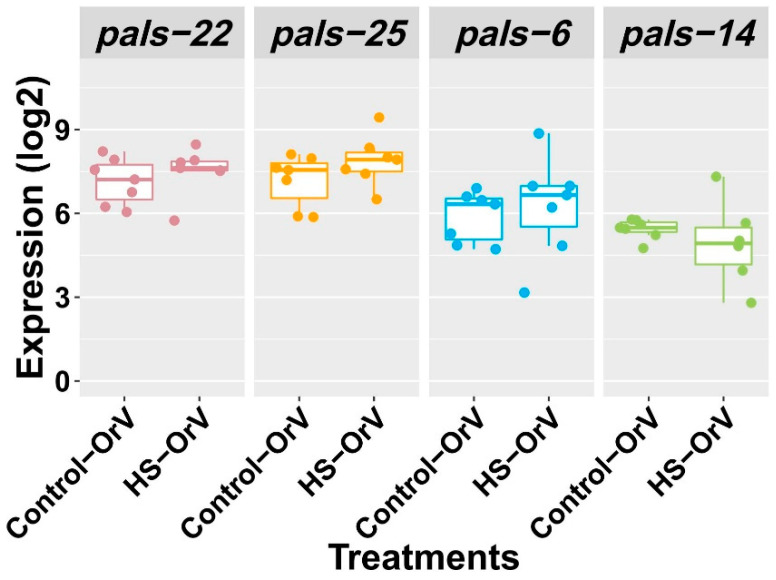
Gene expression of *pals-22, pals-25, pals-6,* and *pals-14* in JU1580 after HS and OrV infection. No significant differences were found between the side-by-side comparisons. Tukey HSD test, *p* > 0.05. Control-OrV: only OrV infection; HS-OrV: HS applied before the OrV infection. Dots: infected biological replicates.

## Data Availability

Microarray data were deposited at ArrayExpress ((E-MTAB-10214). The code used for analysis is shared via a git repository: https://git.wur.nl/published_papers/huang_2021/, accessed on 16 March 2021.

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
