# Peer review of "Heat Stress Reduces the Susceptibility of Caenorhabditis elegans to Orsay Virus Infection"

_genes, 2021, doi:10.3390/genes12081161_

Round 1

Reviewer 1 Report

In this paper the authors try to address if there is an interaction between the level of viral infection seen in a strain and that strains exposure to the abiotic stress of increased temperature.  This make sense as an interaction to dissect as there is evidence for the transcriptional pathways that are activated by these two types of stress to overlap.  However, there are some substantial concerns about both the rigor of some of the experiments and the writing of the paper.  My major concerns our summarized here:

  • Even though the number of replicates for the experiments have been increased- it is clear from the figures that for some of the experiments the number of replicates is fewer than the authors suggest it is in the methods. Given the apparent variation in viral load from experiment to experiment, it is really important that sufficient replication is done to determine if the trends being seen are correct.
  • Although the authors claim that they only did single replicates for their microarray data to find candidate genes – I’ve never encountered publication of expression data done with no replication which is necessary to truly know if results seen are not spurious. The authors do not do any follow up experiments using something like qPCR to validate their findings (they do show one qPCR experiment- however only 2/4 target chosen where from the microarray data and it was not done in either of the strains that were done for the microarray experiment).  Do to the lack of replication and lack of validation of any of the targets this data is preliminary data at best, and not publication ready.
  • The writing of the paper is poor. There are many examples of general poor grammar.  More importantly there are many places where data is not properly or confusingly explained, where previous research is not clearly explained leading to confusion of the goals or reasons behind experiments, or where different terms are used to describe the same thing.  There needs to be a large amount of editing to make this paper clear and accurate.  In addition, there are protocols missing from the methods section that are crucial. 

Below are specific comments on data or writing some major and minor organized by each section of the paper:

Introduction:

Sentence lines 42-43 should be re-written as “The infection by OrV triggers three antiviral mechanisms: the RNA interference (RNAi) response, the uridylation response, and the integrated pathogen response (IPR), an innate transcriptional response.”  The way it is currently written there is not clarity between the two first type of response as separate pathways. 

line 52-53: Should read: To combat the effect of HS, processes like autophagy and the heat-stress response (HRS) maintain proteostasis.    No “the” before proteostasis.

line 54: Should read: HS induces the HSR…

lines 54-56: This sentence is unclear and potentially inaccurate.  HS induces HSF-1 to form oligomers, not prevent them.  Once HSF-1 is oligomerized it translocates to the nucleus, activates the HSR, and then those HSR proteins prevent the formation of misfolded protein oligomers/helps refold misfolded proteins. 

lines 59-62.  Making the leap between the pals-22 mutant phenotype to the statement that IPR and HSR have gene functionality is only possible if the reader knows that pals-22 is a repressor of the IPR pathway.  Knowing this shows that pals-22 represses both thermal tolerance and pathogen tolerance- this needs to be more explicitly laid out in the introduction for the reader to follow.  The previous section where it is stated that pals-22 and pals-25 are antagonistic genes in the IPR pathway does not clearly lay out which of the two partners in this antagonistic relationship promotes and which represses the IRP pathway.

Materials and Methods:

The preferred nomenclature for strain names is to give the strain name first- followed by the genotype.  Such as RB1330 npr-1(ok1447).

lines 75-77 and 95: For most synchronized experiments worms are bleached, eggs are collected, and then hatched in the absence of food.  It is unclear here, especial for experiments described on line 95, but it seems like the authors bleached- and then moved the eggs directly to food for 20 hrs.  This maters because it also determines what stage the treatments are done at- 20hr at 20C without starving would be an L1 worm, while with starving would be an L2.  Finally it would be nice for the authors to clearly state what stage of worms were both treated with the different stressors and also what stage worms were at when collected for analysis.  It seems like depending on the experiment that was L4 or Adult but I’m not sure.

line 92: The method for viral infection is not clear enough.  The citation given is a PhD thesis (Sterken 2016)- and no indication of where within the thesis the protocol should be found is given.  Some brief details of the protocol should be given here and either a page number for the thesis or a different citation (perhaps reference 13 Sterken et al., 2014) should be used instead.

Results:

The title of the first section of the results seems to not capture the data correctly- it seems like the short time period (3 hours) between HS and infection is the key but that isn’t what was shown in the specific section and only works if one includes information from subsequent sections of the paper. 

Line 247- “…and age of HS”.  Should be changed to “… and the age of the worm when it is exposed to HS.” 

Figure 2: It would be helpful if the order of the treatments in the model (Fig 2A) and the graph (Fig 2B) were the same- like in Figure 1- where the model had treatments going top-to-bottom and the figure had treatments going left-to-right.

The type of fluorescent beads used and the source should be present in the MM- not just listed as a reference.

There needs to be more detail in the Methods on the Microarray data analysis.  For example, for the correlation analysis what were the authors correlating- raw intensities from each technical replicate? normalized and averaged intensities?  What type of correlation analysis was done?  In the cor function it could be a Pearson, Spearman or Kendall.

Lines 300-301: On these lines the authors write “Interestingly, in CB4856 strong changes in expression were observed depending on the application of HS, whereas in N2 such transcriptional changes were less apparent (Figure 3a).”   This isn’t clearly written- what I assume the authors are indicating is that in CB4856 there are a subset of genes that appear to be upregulated upon OrV exposure only when the worm is heat shocked and second subset of genes that are upregulated upon OrV exposure which are upregulated weather or not the worm is heat shocked, while in N2 there are no genes whose upregulation upon OrV infection only occurs when the worm is heat shocked.  However, since the dot plots in Fig 3 are not clearly explained and the sentence isn’t very clear, it took me a long time of staring at the graph to figure this out.  This section needs to be re-written for clarity.

Line 330: it should be clarified that in this context the authors mean that pals-22 and pals-25 act antagonistically to each other

Line 331-332: the authors say that knock-out mutant in both genes play a role in thermotolerance, but then only say what the result of the loss of pals-22 is and not pals-25 which given the antagonistic relationship between then would be expected to have the opposite phenotype.  In addition, the authors need to lay out the logic of why they would do the double mutant and what the expectations of these experiments would be given the previous data known on these mutants.  

Figures 4-6: mutant or gene names at the top or bottom of graphs should be italicized per normal C. elegans nomenclature

In the figure legends and the MM the authors indicate number of replicates that were used per each experiment.  However, there are not always the same number of dots on the figure as the authors say there are replicates.  In the MM they indicate that if there was no infection they indicated they left that worm out of the analysis, thus this may be why there are fewer dots.  However, since those worms were completely left out of the analysis they should not be counted towards the number of biological replicates.

When discussing the microarray experiment the authors need to consistently call the different treatments the same thing across writing and different figures.  For example in Figure 3A the treatments are called OrV, mock control, and mock heat-shock, in Figure 3B the treatments are called CT OrV, HS OrV, and mock (not distinguishing between the two mock experiments), and in Figure S2A the treatments are called CT:inf, CT:mock, HS:inf, HS:mock.  It takes a lot of work on the part of the reader to rethink in each figure which experiment the authors are discussing with a new designation each time.

The justification for investigating pals-22 and pals-25 seems weak- the authors  note that “many pals-genes are among the 59 genes” with changes in expression after OrV infection- they fail to note that of the 13 pals-genes in this group- none are pals-22 nor pals-25.  The pals-genes exist in clusters in the genome and the specific cluster of pals- genes tend to be expressed together (Leyva-Diaz et al. (2017) Genetics. 207:529-545).  pals-22 and pals-25 exist in a cluster with pals-23 and pals-24.  None of these four genes are misexpressed in the experiments presented here- thus the jump to look at these particular experiments is not logical versus looking at the pals-genes that were misexpressed.

Lines 349-350:  The authors say that the double mutant display the wild-type phenotype and that they “had a similar sensitivity” as N2- I’m assuming that what the authors mean is that the level of the viral load of the double mutant was not significantly different than N2 under any of the treatment conditions and that like N2 the double mutant showed decreased infection when the strain was also treated with heat shock.  But the way the authors wrote this wasn’t fully clearly and requires the reader to make the jump to assume both the statements are what they mean.  Also, since the figure is broken up by treatment and doesn’t directly address how the different treatments changed the level of infection within a specific strain- the authors need to write clearly and directly if there were significant differences in the level of infection between the treatments (for example- given the spread in the data of the pals-22/25 in the HS-OrV treatment it is NOT clear if there is a significant difference in viral load between that and the control. 

Line 356: what does “with the nearest location” mean when referring to the hsp16.48?  Does it mean that it is the gene with a mutation closest to it?  Or that the strain with the mutation in hsp-16.48 is found in a lab or resource center close to the lab?  If the former- it is confusing the write it this way, if the later this information isn’t necessary and should be removed.  It makes it sound like there are mutations in the other hsp-genes but the authors didn’t use them because they are not housed in locations close to their lab.  Since worm strains can be sent internationally this isn’t a reason to not get a strain.

Discussion:

Lines 408-410:  The authors comment that they saw different things in figure 1 vs figure 4 with the effect under HS-OrV in N2 and attribute this to potentially having fewer replicates in Figure 4.  There are two problems with this statement 1) before this point the authors never directly address what the differences are between the N2 treatments in figure 4 which does not directly show the statistical differences between treatments within a strain (and is set up to not focus on this by having different treatments on different graphs), 2) IF the authors had done the number of replicates for figure 4 as they indicated they should have done for figure 4 (namely 7) than understanding if this difference was due to random sampling/experimental differences as compared to differences in the underlying biology would be clearer, instead they only have 5 replicates for figure 4.  The authors in general fail to explain why in some instances (like the Con-OrV and OrV-HS treatments) the spread of the data in N2 is much tighter than in other instances.  This makes it very hard to know if the non-significance seen in some of their other experiments- like those for JU1580 in figure 5 are due to bad data or real biology. 

Lines 435-437: This sentence is really unclear- and doesn’t accurately reflect the paper it is citing. 

Overall, the discussion feels very disjointed and the topics do not flow together.  It is hard to determine what the authors feel the main take home from their experiments.  This section also shows the least polish in the writing which also makes it harder to follow.

Author Response

We thank the reviewer for the detailed comments. Please see the attachment for our response.

Reviewer 2 Report

The inclusion of additional replicates strengthens the authors conclusions and my other concerns have been addressed.

Author Response

We thank the reviewer for this.

This manuscript is a resubmission of an earlier submission. The following is a list of the peer review reports and author responses from that submission.

Round 1

Reviewer 1 Report

Animals contend with many types of biotic and abiotic stresses, including temperature and viruses. Often animals are exposed to multiple stresses at the same time, but there is not a lot known about the interplay between these different types of stresses.  Here, the authors use C. elegans and its naturally occurring Orsay virus to investigate the effects that temperature has on viral infection. The authors show that heat stress delivered either before or after viral infection reduces viral load. The authors used explorative microarray analysis on a single replicate to show that the two stresses had similar expression. The authors then tested several mutants, showing that heat stress caused decreased viral levels in pals-22/pals-25 and several heat shock mutants, demonstrating that these pathways are not necessary for the response. Together, the data suggests that heat stress and viral infection upregulate a similar set of genes, and that heat stress can reduce viral infection.

Major point:

  1. Although some statistics were performed, it is difficult to interpret the data as these tests are not included on the figures. All the statistical tests for viral infection should be included on the figures and the tests and p-values stated in the figure legends.

2.In Figure 4, heat shock then Orsay no longer shows a difference in N2. Why is this?

Minor points:

  1. Y-axis label on figure 2b is confusing. Replace food intake index with something like “% fluorescence”
  2. Methods line 85:, The authors state that jy1 and jy11 were generated using crispr, but in fact they were generated with forward genetic screens, not crispr (Reddy et al plos pathogens 2019). This needs to be corrected.
  3. Line 76. MQ needs to be defined.
  4. 176 to 177. Ii is not clear how the feeding assay was performed. Were 8 plates set up per condition and a single animal was analyzed from each condition, or was only a single plate per condition set up and 8 animals analyzed? This needs to be clarified.
  5. For the viral infection and microarray experiments the number of animals per plate in each condition needs to be stated.
  6. Should cite PMID: 31034475 which shows that increased temperature has been reported to induce lasting immunity to bacterial infection in C. elegans.

Reviewer 2 Report

Huang and colleagues submit a manuscript addressing the interplay between biotic and abiotic factors in animal health, in this case the interaction between viral infection and heat stress, both known to impact proteostasis. They find that heatshock can have a protective effect for C. elegans – decreasing viral load if heatshock is applied shortly before or after viral challenge. They then go on to try to look at the signaling pathways that may be part of the interaction between heat stress and viral challenge, based on candidates selected from preliminary microarray data. Figure 1 looks solid, and perhaps Figure 2. However, a combination of high variability and low N make it impossible to draw any conclusions from the latter part of the manuscript (Figures 3-6). Put another way, the data is not sufficient to support the conclusions. The research question is an interesting one, and I wish the authors luck in their future work. Specific comments below:

Major comments:

For the data in Figure 4-6, the n is far too low and the variability too high to draw any conclusions. In most cases it’s an n of 3, sometimes it appears to be 2, or even 1. The authors are aware of this, as they note that the low n combined with high variability preclude making sound conclusions when they discuss the differences between the N2 data in Figure 1b and Figure 4 (L408).

One example to illustrate: L347 “Contrary to N2, HS (OrV-HS) does not 349 cause a significant decrease in viral load for the pals-22 mutant.” – you can’t tell, because the pals-22 data is an n of 1.

The presentation of the data in Figures 4-6 is strange, and a bit misleading. Box plots would normally be used to display data with a much higher n, but here they are used for an n’s of 3, 2 and 1. The places where it is 1 (e.g. Figure 4, pals22, OrV-HS) the plotting software displays the datapoint as the upper bound, then the median and lower bound as 0, and when there are two it assigns the upper bound and median to the two data points, and the lower bound as 0. This is confusing. There’s also no reason to highlight the median when there are only 1-3 data points.

Appropriate statistical tests need to be used throughout. For example t tests are not suitable when an experiment contains more than two groups (L349), because you get an increased chance of false positives with each comparison. One-way ANOVA with an appropriate post-test is commonly used by biologists when more than one group is involved.

It’s strange to do the microarray in one strain, and the rest of the experiments in others. Also, as the authors note, it’s an n of 1. Some comfort can be derived from at least having technical replicates, and pulling out a list of genes that make sense.

Minor comments:

There are several instances of C. elegans not being italicized throughout the manuscript

L59: “This can be illustrated by the mutation of pals-22” – it would be helpful for readers to clarify loss of function vs gain of function

L75: E. coli should be italicized

L76: is “MQ” Milli-Q?

L80-85: all alleles should be italicized

L164: should cite the FIJI paper – Schneiderlin et al. (2012) Nature Methods 9(7): 676-682